# Extracts from European Propolises as Potent Tyrosinase Inhibitors

**DOI:** 10.3390/molecules28010055

**Published:** 2022-12-21

**Authors:** Jarosław Widelski, Katarzyna Gaweł-Bęben, Karolina Czech, Emil Paluch, Olga Bortkiewicz, Solomiia Kozachok, Tomasz Mroczek, Piotr Okińczyc

**Affiliations:** 1Department of Pharmacognosy with Medicinal Plants Garden, Medical University of Lublin, ul. Chodźki 1 (Collegium Universum), 20-093 Lublin, Poland; 2Department of Cosmetology, University of Information Technology and Management in Rzeszów, 2 Sucharskiego, 35-225 Rzeszów, Poland; 3Department of Microbiology, Faculty of Medicine, Wroclaw Medical University, Tytusa Chałubińskiego 4, 50-376 Wrocław, Poland; 4Department of Biochemistry and Crop Quality, Institute of Soil Science and Plant Cultivation—State Research Institute, Czartoryskich 8, 24-100 Puławy, Poland; 5Department of Natural Products Chemistry, Medical University of Lublin, 20-093 Lublin, Poland; 6Department of Pharmacognosy and Herbal Medicines, Wrocław Medical University, Borowska 211a, 50-556 Wrocław, Poland

**Keywords:** propolis, tyrosinase, inhibitor, UHPLC–DAD–MS/MS, antioxidant, melanin, monophenolase, diphenolase, phenolic acid glycerides, galangin, chrysin

## Abstract

Tyrosinase is a key enzyme in the melanogenesis pathway. Melanin, the product of this process, is the main pigment of the human skin and a major protection factor against harmful ultraviolet radiation (UVR). Increased melanin synthesis due to tyrosinase hyperactivity can cause hyperpigmentation disorders, which in consequence causes freckles, age spots, melasma, or postinflammatory hyperpigmentation. Tyrosinase overproduction and hyperactivity are triggered by the ageing processes and skin inflammation as a result of oxidative stress. Therefore, the control of tyrosinase activity is the main goal of the prevention and treatment of pigmentation disorders. Natural products, especially propolis, according to their phytochemical profile abundant in polyphenols, is a very rich resource of new potential tyrosinase inhibitors. Therefore, this study focused on the assessment of the tyrosinase inhibitory potential of six extracts obtained from the European propolis samples of various origins. The results showed the potent inhibitory activity of all tested propolis extracts towards commercially available mushroom tyrosinase. The four most active propolis extracts showed inhibitory activity in the range of 86.66–93.25%. Apart from the evaluation of the tyrosinase inhibition, the performed research included UHPLC–DAD–MS/MS (ultra high performance liquid chromatography coupled with diode array detection and tandem mass spectrometry) phytochemical profiling as well as antioxidant activity assessment using the 2,2-diphenyl-1-picrylhydrazyl (DPPH) and the 2,2”-azino-*bis*(3-ethylbenzothiazoline-6-sulfuric acid (ABTS) radical scavenging tests. Moreover, statistical analysis was used to correlate the tyrosinase inhibitory and antioxidant activities of propolis extracts with their phytochemical composition. To summarise, the results of our research showed that tested propolis extracts could be used for skin cosmeceutical and medical applications.

## 1. Introduction

Propolis, also known as bee glue, is a resinous material, due to its physical properties, used by bees in the hive mainly as a sealing and building material to strengthen the edges of the combs and the entrance to the hive. Bees coat the cavities, damages, and gaps in the walls of the hive with propolis [1,2]. Chemically, propolis is a mixture of plant resin (60%), wax (30%), and volatiles (10%), completed with 5% of other organic compounds [3]. Apart from its well-established antimicrobial properties and antiviral potential, propolis shows numerous biological activities, such as antioxidant, antiparasitic, anti-inflammatory, and finally, anticancer potential (cytostatic activity and ability to prevent tumour growth) [4,5]. Flavonoids, phenolic acids and their numerous derivatives, which belong to polyphenolic compounds, are mainly responsible for a broad spectrum of biological activities of propolis and its extracts [6].

Of particular importance is the ability of propolis to inhibit the activity of various enzymes (acetylcholinesterase, lipoxygenase, α-glucosidase, xanthine oxidase, hyaluronidase), which makes it an important ingredient of dietary supplements and cosmeceuticals, and in the future, a promising source of new active compounds used in the therapies of skin disorders, civilisation, and neurodegenerative diseases [7].

Tyrosinase (EC 1.14.18.1), also known as polyphenol oxidase, is a copper-containing enzyme responsible for the catalysis of the first two rate-limiting steps of melanin synthesis (melanogenesis) [8]. The two tyrosinase-dependent reactions are the hydroxylation of L-tyrosine by monophenolase activity (monophenol L-tyrosine to diphenol 3,4-dihydroxyphenylalanine, L-DOPA), followed by oxidation of L-DOPA to the *o*-dopaquinone (diphenolase activity) [8,9]. Melanin, biosynthesised by tyrosinase in melanocytes, is one of the most widely occurring pigments in nature and has been found in bacteria, fungi, higher plants, and animals, including humans. The main role of melanin, apart from pigmentation, is to protect the skin from harmful factors, such as UVR, by absorbing UV rays and removing reactive oxygen species (ROS) [8,10].

Melanogenesis is a protective response against harmful stimuli, but a reduction in melanin synthesis is very important in the treatment of hyperpigmented diseases, which include various neurocutaneous diseases such as melasma, age spots, and scars from skin damage. The involvement of melanin in the aetiology of malignant melanoma, which develops in melanocytes, is of great interest [8,11]. Tyrosinase is also involved in the enzymatic browning of food and is considered detrimental to the quality and value of plant-based foods and beverages [12,13]. Therefore, controlling the activity of tyrosinase (by inhibiting it) is an important endeavour in both food processing and preservation [13]. Currently, skin whitening products and blemish removers are commercially available for cosmetic purposes in order to even out the skin tone and lighten hyperpigmentation spots [9]. Most skin whitening products are based on competitive inhibitors of tyrosinase [14].

The main aim of the presented study was to evaluate the tyrosinase inhibitory activity of six European propolis samples of different origins. Since propolis extracts with predetermined qualitative compositions were selected for the study, the second objective of the research was to assess the effect of the propolis components on the inhibition of the tyrosinase enzyme.

## 2. Results and Discussion

### 2.1. Analysis of Propolis Extracts by UHPLC–DAD–MS/MS (Ultra High Performance Liquid Chromatography Coupled with Diode Array Detection and Tandem Mass Spectrometry)

The study involved seven samples of propolis: two from Ukraine (UT and UK2), three from Poland (PL2, PL3, and PLU), one from Greece (GP) and one from Russia (RS). The composition of analysed propolis samples and the methods used for the identification of their active compounds were described in our previous publications [15,16]. For that reason, the composition of the propolis samples was presented only in Appendix A. The most important information obtained from the analysis of the composition of the tested propolis samples is their black poplar (*Populus nigra* L.) and aspen (*P. tremula* L.) origin. PL2 and RS exhibited a predominance of aspen markers (phenolic acid glycerides) over black poplar markers, while the remaining samples mainly contained black poplar markers (flavonoids: galangin, chrysin, pinocembrin, and pinobanksins-3-O-acetate).

### 2.2. Tyrosinase Inhibitory Activity of Tested Extracts of Propolis

The propolis extracts were analysed for their ability to inhibit commercially available mushroom tyrosinase. A mushroom tyrosinase inhibitory assay is the most frequently used method for the determination of the potential skin-lightening activity of natural compounds and plant or bee product extracts. Nonetheless, due to the differences between mushroom and animal tyrosinases concerning their functional as well as structural features, particular compounds and extracts can differ in inhibitory activity against both types of enzymes [17]. However, due to low costs, high throughput, and the short time for analysis, the test using the mushroom type of tyrosinase is still considered the first-choice tool for initial screening studies of novel skin-lightening compounds and extracts.

The tyrosinase inhibitory activity of each propolis sample was tested at three concentrations of 25.00, 50.00, and 100.00 µg/mL (Figure 1). The most significant mushroom monophenolase tyrosinase inhibition activity (Figure 1a) at 100.00 µg/mL was detected for sample UK2 (93,25% of inhibition), followed by samples PLS2, UT, and GP, showing 90.93%, 88.70%, and 86.55% of inhibitory activity, respectively. At the concentration of 50.00 µg/mL, the strongest inhibitory activity was shown for the PLU sample (88.40%) and successively for UT (87.35%), PLS2 (86.37%), UK2 (85.75%), and GP (83.00%) propolis extracts. The activity of five of the most effective propolis extracts (UT, PLS2, PLU, UK2, and GP) at the concentration of 25.00 µg/mL was similar and in the range of 83.15–86.90% (the UT sample showed the highest inhibition level of these five). The least active propolis samples, PLS3 and RS, inhibited the monophenolase activity of tyrosinase, respectively, by 67.04%, 57.03%, 28.74% and 56.03%, 47.13%, 31.27% at the tested concentrations (100.00; 50.00 and 25.00 µg/mL).

In the case of diphenolase mushroom tyrosinase inhibitory activity (Figure 1b) at 100.00 µg/mL, the most active were samples PLS3 (66.88% of inhibition) and RS (66.73%), followed by UT and PLS2 with inhibition at levels of 65.03% and 62.66%, respectively. The rest of the tested propolis samples exhibited weaker diphenolase inhibitory potential: UK2 (58.03%), PLU (52.83%), and GP (36.04%). At the concentration of 50.00 µg/mL, the strongest inhibitory activity was presented by the PLS3 sample (60.84%), followed by the UT (52.21%) and RS (51.35%) samples. The rest of the samples showed tyrosinase inhibitory activity in a range from 21.41 to 47.49%. The activity of the four most effective propolis extracts (RS, PLS2, PLU, UK2, and GP) at a concentration of 25.00 µg/mL was similar and varied from 35.39 to 38.77% of inhibition.

To date, there are only a few scientific reports on the ability of extracts obtained from propolis to inhibit tyrosinase activity. The tyrosinase inhibitory potential was described for twenty propolis samples obtained from different parts of Greece [5]. The results showed the highest activity of samples with predominant terpenoid content (68.46–85.69% of inhibition at 200 µg/mL). Samples rich in flavonoid compounds showed inhibition in the range of 40.14–56.66%, and propolis samples of mixed composition inhibited tyrosinase by 37.24–47.25% [5]. In another study, propolis extracts from the samples acquired from four stingless bee species (*Homotrigona apicalis*, *Wallacetrigona incisa*, *Tetragonula fuscobaleata*, and *Tetragonula fuscibasis*) were screened for melanogenesis regulatory activity [2]. Tested extracts at a concentration of 100 µg/mL showed the inhibition of tyrosinase at a range of 14-53%. The highest inhibitory activity was shown for the extract of propolis produced by the species *Wallacetrigona incisa*, rich in two prenylated flavonoids: broussoflavonol F and glyasperin A [2]. In another work, among 21 samples of propolis collected in different locations in Morocco, two of them exhibited potent inhibitory activity towards tyrosinase, with IC_50_ values of 0.050 and 0.037 mg/mL [7]. Moreover, a negative correlation between IC_50_ and the concentration of polyphenolic compounds (phenols, flavones, and flavanones) was noticed [7]. This activity can be explained, according to El-Guendouz [7] and Fu and co-authors [18], by a mechanism of chelation of the copper ion, Cu^2+^, located at the active site of the tyrosinase enzyme by vicinal 3′,4′-dihydroxyl group (catechin) or the structure of 3-hydroxy-4-keto moiety of some flavonoids.

The results presented that the activity of different types of propolis (characterised by different origins and chemical compositions) in inhibiting tyrosinase only underlines the great potential of the propolis extracts we studied as natural inhibitors of this enzyme. The samples of European propolis with a high content of polyphenolic compounds tested in this study were selected for analysis due to the presence of marker compounds indicating the origin of poplar, aspen–birch, aspen–poplar, and aspen–birch–poplar. It seems that this chemical composition has an impact on the outstanding biological activity of propolis, including the ability to inhibit tyrosinase strongly. In addition, compared to other results on the level of tyrosinase inhibition, the propolis samples tested by our group showed stronger inhibition of tyrosinase than most of the propolis and plant extracts at a lower dose tested by others. For example, in the studies conducted on 92 herbal and propolis extracts by Deniz and collaborators [19], only 12 investigated samples showed a tyrosinase inhibitory effect higher than 50% at a relatively high concentration of 666 µg/mL. Very promising for future research is also the fact that our propolis samples manifested activity similar to reference substances (kojic acid) used in these studies [19].

### 2.3. Antioxidant Activity of Tested Propolis Extracts

Propolis extracts are well known as very rich natural mixtures showing significant antioxidant activity resulting from the presence of various types of polyphenolic substances [20,21]. In this study, the assessment of the antioxidant activity of propolis samples was performed using the 2,2-diphenyl-1-picryllhydrazyl (DPPH) and the 2,2″-azino-*bis*(3-ethylbenzothiazoline-6-sulfuric acid (ABTS) free radical scavenging assays (Table 1). All samples exerted significant antioxidant activity. The results obtained using the DPPH scavenging test were diverse between tested samples, with EC_50_ values between 9.34 and 32.22 µg/mL. In this test, the highest antioxidant potential was observed for UK2 and PL3 samples, whereas the least active was sample PLU. In the ABTS scavenging assay, the results were less varied, showing the EC_50_ values from 2.12 to 2.47 µg/mL, with one exception, 5.12 µg/mL for the least propolis sample from Greece (GP).

### 2.4. Impact of Antioxidant Activity of Propolis Extracts and Their Components on Tyrosinase Inhibitory Activity

The tyrosinase inhibitory activity of propolis extracts was tested in two experimental models—the reaction of tyrosinase and L-tyrosine as its substrate (monophenolase activity model) or tyrosinase and L-DOPA as substrate (diphenolase activity model). The obtained results were also correlated with the values from the antioxidant activity tests. A single-component correlation matrix model was also performed. The raw results are presented in Appendix A.

Based on obtained correlation values, it might be concluded that the monophenolase inhibitory activity is not correlated with the results of the antioxidant colorimetric tests, while a positive correlation was found between ABTS results and diphenolase inhibitory activity for all extracts’ concentrations. A different result was obtained in the literature, where diphenolase activity was not correlated with other values [5].

When single components of analysed extracts were correlated with antioxidant test results, positive and negative correlations were found with some of the compounds.

In the case of diphenolase activity, mostly a negative correlation (or no correlation) with single components was calculated. More complex results were obtained for monophenolase inhibitory activity—except for some minor negative correlations, only one positive correlation between diphenolase inhibitory activity and galangin-7-methyl ether content was observed, which increased with the extract concentration. It is generally known that different flavonoids and their aglycones act as competitive, non-competitive, or mixed inhibitors of tyrosinase [12,13,22]. Poplar propolis usually contains many flavonoids known from the literature as effective inhibitors of tyrosinase activity, such as galangin [12,13,22], chrysin [13], pinocembrin [23], pinobanksin-3-*O*-acetate [23], and others. It is significant that these flavonoids usually work better on the monophenolase inhibitory activity model than the diphenolase inhibitory assay [22]. This fact may be explained by the lack of hydroxyl groups in positions 3′ and 4′ of the flavonoid B-ring. The lack of these groups is responsible for worse solubility in water of these components and probably weaker activity in the diphenolase models or activity unable to be determined [24].

In comparative research by Fan et al. [13], it was shown that some of the primary propolis flavonoids (galangin and apigenin) exhibited a strong inhibitory effect in the monophenolase activity model, while another (naringenin) was significantly weaker. Moreover, it was proven that the relationship between the flavonoid structure and the tyrosinase inhibitory activity is very complex [13]. For example, it was shown that methylation usually decreases the inhibitory activity of flavonoids, but not in all cases. Interestingly, in our research, the correlation between the tyrosinase inhibitory activity and the content of the methylated flavonoid (galangin-7-methyl ether) was calculated. Therefore, propolis is a complex matrix, and the activity of whole extracts depends on the interaction between different components [25]; further research is required in the future to explain this phenomenon. A potentially observed correlation may be random and results from the presence of a specific mix of components. In another research, it was shown that pinobanksin-3-acetate and pinocembrin exhibited 35% and 24% inhibition of tyrosinase activity in the monophenolase model, respectively, at the concentration of 200 µg/mL [23]. At the same concentration, pinocembrin inhibited only 16.78% of tyrosinase activity in the diphenolase model [9]. In our research, whole propolis extracts exhibited even above 80% of the monophenolase inhibitory activity and above 30% of the diphenolase inhibitory activity at a concentration of 25 µg/mL. In our opinion, it is possible that some components of propolis may increase the solubility of other components in water, which may increase their activity. On the other hand, components in extracts interact with different domains of tyrosinase at the same time and block its activity by different mechanisms. For that reason, the relatively weak tyrosinase inhibitory activity of single components may result in a significantly stronger tyrosinase inhibitory activity of their mixture. However, this hypothesis regarding the propolis extracts requires further research.

The principal component analysis (Figure 2 and Figure 3) exhibited that 81.73% of variability may be explained in a two-factor model. Factor 1 explained 60.06%, while factor 2 explained 21.67% of the variability. Samples may be divided into three groups in Figure 3. The observed division was not accurately corresponding with the plant origin or the biological activities.

## 3. Materials and Methods

### 3.1. Propolis Samples

Propolis samples used in this study have been summarised in Table 2 below.

The obtained propolis was frozen in liquid nitrogen and crushed in a mortar. The procedure was repeated three times. Propolis samples included in this research were also used in our previous studies [15,16], and therefore, their abbreviations were kept the same.

### 3.2. Extraction of Propolis Samples

Propolis samples were grounded and extracted using ethanol in water (70:30; *v/v*) in a proportion of 1.0 g of propolis per 10 mL of the solvent. The extraction was performed in an ultrasonic bath (Sonorex, Bandelin, Germany). The extraction conditions were set at 40 °C for 45 min and 756 W (90% of the ultrasound bath power). The obtained extracts were stored at room temperature for 12 h and then filtered through Wattman No. 10 filter paper.

### 3.3. UHPLC–DAD–MS/MS Analysis of Propolis Extracts

The composition of propolis extracts was analysed by the Waters Acquity UPLC system (Waters, Milford, CT, USA) equipped with PDA 200–500 nm, a mass spectrometer Xevo-Q-TOF (Waters, Milford, CT, USA), and a column BEH C18 130 Å (1.7 μm, 2.1 mm × 150 mm) (Waters, Milford, CT, USA). Analyses were performed according to previously described methods [15,16].

### 3.4. Mushroom Tyrosinase Inhibitory Assay

The tyrosinase inhibitory activity of propolis extracts was analysed using L-tyrosine and 3,4-dihydroxy-L-phenylalanine (L-DOPA) as substrates in order to investigate the monophenolase and diphenolase inhibitory activity, respectively [26,27]. The monophenolase inhibitory assay was performed by mixing 80 µL of 100 mM phosphate buffer (pH 6.8), 20 µL of propolis extracts (1 mg/mL, 0.5 mg/mL, or 0.25 mg/mL) or solvent control, and 20 µL of tyrosinase working solution (500 U/mL), followed by 10 min pre-incubation at RT. After this time, 80 µL of 2 mM L-tyrosine was added to each sample. For the diphenolase inhibitory assay, 120 µL of 100 mM phosphate buffer (pH 6.8) was mixed with 20 µL of propolis extracts (1 mg/mL, 0.5 mg/mL, or 0.25 mg/mL) or solvent control, and 20 µL of tyrosinase working solution (500 U/mL). The samples were pre-incubated for 10 min at RT and mixed with 40 µL of 4 mM L-DOPA. For both the monophenolase and diphenolase inhibitory assays, the absorbance of a formed dopaquinone was measured at λ = 450 nm following 20 min of incubation at RT in darkness using the FilterMax F5 microplate reader (Molecular Devices, San Jose, CA, USA). The values were corrected by the absorbance of the diluted propolis extracts without tyrosinase, L-tyrosine, and L-DOPA. A control sample (100% tyrosinase activity) contained phosphate buffer, tyrosinase, an equal volume of the solvent, and the appropriate dose of each substrate. The monophenolase or diphenolase inhibitory activity was calculated based on the equation:% of inhibitory activity = 100% − [(AbsS/AbsC) × 100%](1)
where AbsS—the absorbance of the sample (extract + tyrosinase + substrate) and AbsC—the absorbance of the control sample (solvent + tyrosinase + substrate).

### 3.5. Antioxidant Activity Assay

The antioxidant activity of the propolis extracts was analysed by the DPPH and the ABTS radical scavenging assays, as described by Matejic et al. and Re et al., respectively [28,29]. In the DPPH radical scavenging assay, 100 μL of diluted propolis extracts or L-ascorbic acid (1000–0.98 µg/mL) was mixed with 100 μL DPPH working solution (25 mM in 99.9% methanol; A_540_ ≈ 1). An amount of 100 μL of methanol solvent was mixed with 100 μL DPPH and was used as a control sample (100% radical activity). Following 10 min of incubation at room temperature in darkness, the absorbance of the samples was measured at λ = 540 nm using a microplate reader (FilterMax F5 Molecular Devices, San Jose, CA, USA). For the ABTS radical scavenging assay, 7 mM ABTS solution in 2.45 mM K_2_S_2_O_8_ was prepared and diluted in distilled H_2_O in order to obtain the ABTS radical working solution (A_405_ ≈ 1). Fifteen μL of diluted propolis extracts or L-ascorbic acid (2000–1.95 µg/mL) was mixed with 135 μL ABTS working solution. Fifteen μL of methanol solvent mixed with 135 μL ABTS was used as a control sample. After 15 min of incubation at room temperature in darkness, the absorbance of the samples was measured at λ = 405 nm using a microplate reader. The percentage of radical scavenging in both assays was calculated for each sample based on the following equation:% of radical scavenging = [1 − (Abs(S)/Abs(C))] × 100(2)
where Abs(S)—the corrected absorbance of the extract and Abs(C)—the corrected absorbance of the control sample (radical + solvent).

The IC_50_ value was defined as the concentration of dried extract in µg/mL that is required to scavenge 50% of the radical activity and calculated using GraphPad Prism 9.0 software (GraphPad Software, San Diego, CA, USA).

### 3.6. Statistical Analysis

Statistical analysis was performed using Statistica 14.0.0.5 software (Tibco Software Inc., Palo Alto, CA, USA). The correlation between the composition and the biological activities of tested propolis extracts was performed using a correlation matrix. The matrix was composed of the percentage of UV chromatograms (200–500 nm), the relative peak area, total phenolic content (TOP), flavonoid content (TF), and tyrosinase inhibition values. Substances of at least 1% of the relative area (in any sample) were used to construct the matrix. During the analysis, r, p, and N values were calculated. The prepared matrix was attached in Appendix A (Appendix A. Statistical input data).

## 4. Conclusions

The conducted research showed a very strong inhibitory activity of European propolis extracts on the tyrosinase enzyme and the melanogenesis process itself. The complexity of propolis is reflected in the polyphenol-rich chemical profile of the tested extracts. Their interesting, potent, and multi-directional biological activity, presented in our previous studies, encourages further experiments with fractionation and the targeted isolation of biologically active compounds. Moreover, due to the presence of numerous natural polyphenolic compounds in propolis, which have antioxidant, antimicrobial, and anti-tyrosinase activities, this natural product can be successfully used in the cosmeceuticals industry and food supplements, affecting the condition of the skin and for the treatment of various skin disorders.

## Figures and Tables

**Figure 1 molecules-28-00055-f001:**
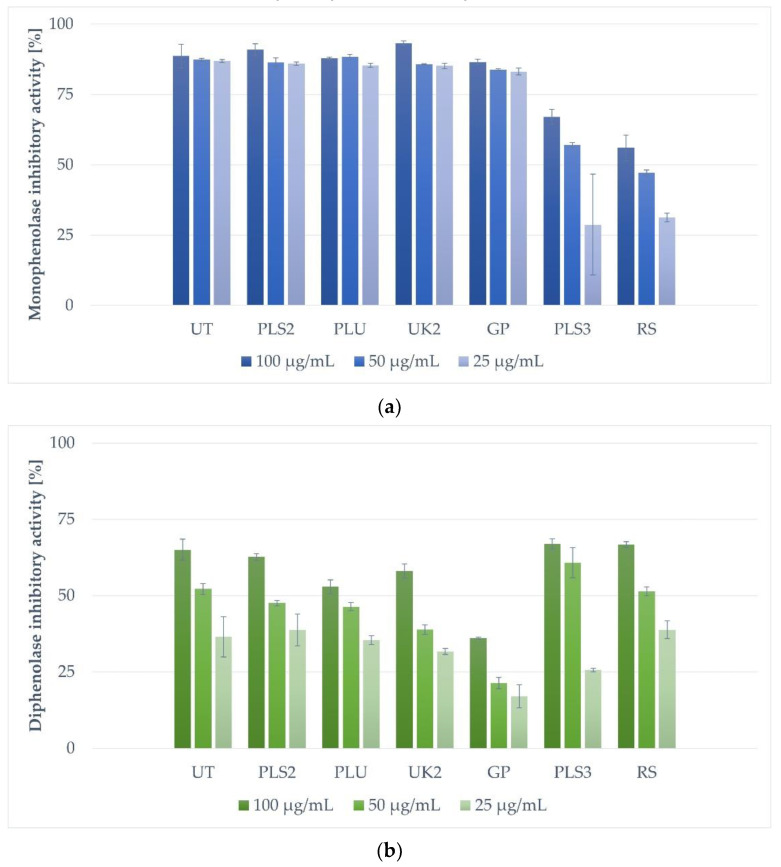
Inhibition of monophenolase (**a**) and diphenolase (**b**) by tested propolis extracts. Table legend: UK2—Ukraine, Khmelnitsky Village; UT—Ukraine, Tarnopol; PLS2—Poland, Lower Silesia; PLS3—Poland, Lower Silesia; PLU—Poland, Lublin Region; GP—Greece, Parga; RS—Russia, Saratov Oblast.

**Figure 2 molecules-28-00055-f002:**
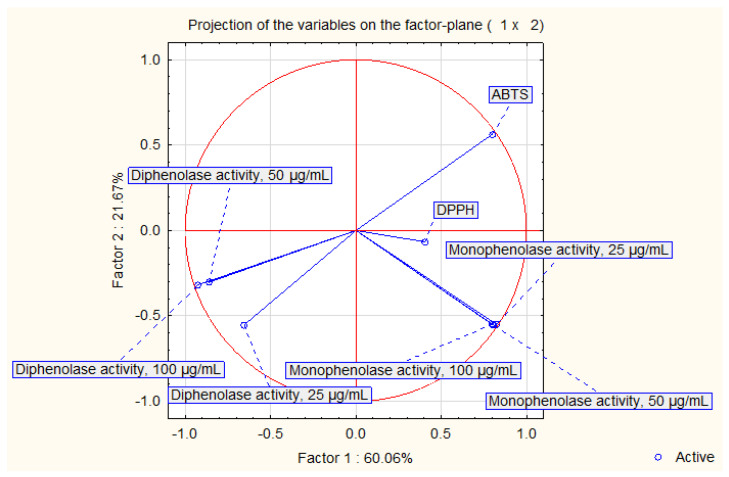
Projection of the variables on the factor plane in a two-component model.

**Figure 3 molecules-28-00055-f003:**
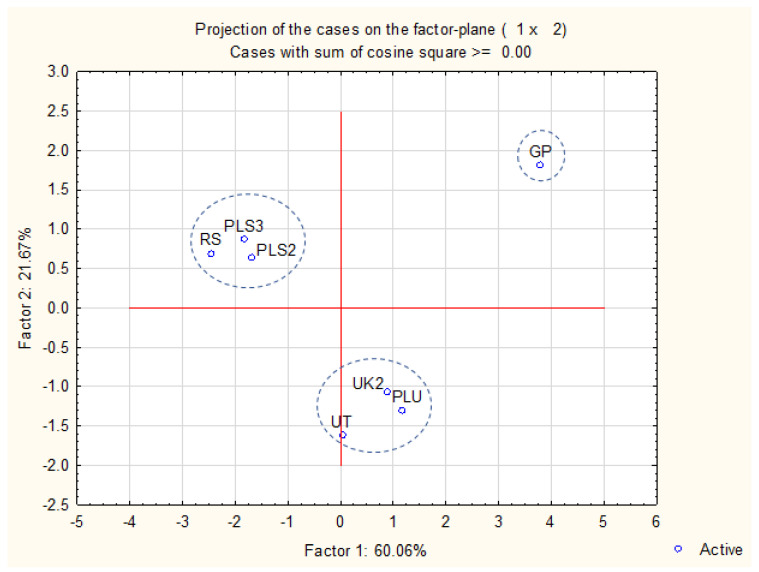
Projection of the cases on the factor plane in a two-component model.

**Table 1 molecules-28-00055-t001:** Antioxidant activity of tested propolis extracts.

	EC50 (µg/mL)	
	DPPH Scavenging	ABTS scavenging
UT	11.16 ± 0.54	2.47 ± 0.27
PLS2	19.05 ± 0.54	2.41 ± 0.49
PLU	32.22 ± 0.36	2.43 ± 0.32
UK2	9.85 ± 0.26	2.37 ± 0.43
GP	20.50 ± 0.34	5.12 ± 0.18
PLS3	9.34 ± 0.22	2.43 ± 0.43
RS	13.48 ± 0.57	2.12 ± 0.29
Vit C	1.45 ± 0.29	0.70 ± 0.02

Table legend: UK2—Ukraine, Khmelnitsky Village; UT—Ukraine, Ternopil; PLS2—Poland, Lower Silesia; PLS3—Poland, Lower Silesia; PLU—Poland, Lublin Region; GP—Greece, Parga; RS—Russia, Saratov Oblast; Vit C—vitamin C.

**Table 2 molecules-28-00055-t002:** Origin and abbreviations of analysed propolis samples.

Abbreviation	Country	State/Region
UK2	Ukraine	Khmelnitsky Village
UT	Ukraine	Ternopil
PLS2	Poland	Lower Silesia
PLS3	Poland	Lower Silesia
PLU	Poland	Lublin
GP	Greece	Parga
RS	Russia	Saratov Oblast

## Data Availability

Research data are available from the authors.

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
