# Peer review of "Extracts from European Propolises as Potent Tyrosinase Inhibitors"

_molecules, 2022, doi:10.3390/molecules28010055_

Round 1
Reviewer 1 Report
This manuscript evaluated tyrosinase inhibitory potential of six extracts obtained from the European propolis samples of different origin. uHPLC-DAD-MS/MS phytochemical profiling and antioxidant DPPH and FRAP tests were carried out. The authors used statistical analyze to searching connection between tyrosinase inhibition and extracts composition as well as antioxidant activity. The content is suitable for the readers of the journal. Overall, the quality of data presented is in good manner and scientifically acceptable. I think this paper need a minor revision before it could be accepted to publish.
I have following suggestions for revision:
1. There are minor grammar and print errors in the text, please check up throughout the manuscript.
2. Please check the referencing format as well.
Specific comments for the authors:
Line 32 Delete the extra Period.
Line 33 Please check the grammar.
Line 114-115 58,03 52,83 36,04 correct these numbers
Line 118 similar , delete space
References 12-18, 28-29 Please check the referencing format.
Author Response
Response to Reviewer #1
This manuscript evaluated tyrosinase inhibitory potential of six extracts obtained from the European propolis samples of different origin. uHPLC-DAD-MS/MS phytochemical profiling and antioxidant DPPH and FRAP tests were carried out. The authors used statistical analyze to searching connection between tyrosinase inhibition and extracts composition as well as antioxidant activity. The content is suitable for the readers of the journal. Overall, the quality of data presented is in good manner and scientifically acceptable. I think this paper need a minor revision before it could be accepted to publish.
We would like to thank the Reviewer for his/her time and valuable comments aiming to improve the quality of our Manuscript. Please see below the responses to your particular suggestions, indicated in green.
I have following suggestions for revision:
- There are minor grammar and print errors in the text, please check up throughout the manuscript.
Response: The grammar and print errors have been corrected through the whole manuscript.
- Please check the referencing format as well.
Response: The format of the references have been corrected.
Specific comments for the authors:
Line 32 Delete the extra Period.
Response: Correction has been done.
Line 33 Please check the grammar.
Response: The sentence has been corrected.
Line 114-115 58,03 52,83 36,04 correct these numbers
Response: The numbers have been corrected.
Line 118 similar , delete space
Response: Correction has been done.
References 12-18, 28-29 Please check the referencing format.
Response: The references have been corrected based on the Molecules journal template.
Reviewer 2 Report
1. This article has many grammatical mistakes, even in the title! There are missing verbs, and some sentences are incomprehensible to the readers. Please send this article for proofreading before publication. Please note the differences between US/UK spelling.
e.g
line 57: "of"
line 85: "input"
line 138: Morocco
line 153: another result
line 235: ?
2. There is a jump between numbering (2) Results and Discussion and (4) Materials and Methods.
3. Section 4.2 was copied from one of the authors' other publications.
4. The abbreviations were not explained properly in the "Results and Discussion" because the "Materials and Methods" was placed at the end of the article.
5. There are missing references: line 272.
Author Response
Response to Reviewer #2
We would like to thank the Reviewer for his/her time and valuable comments aiming to improve the quality of our Manuscript. Please see below the responses to your particular suggestions, indicated in green.
- This article has many grammatical mistakes, even in the title! There are missing verbs, and some sentences are incomprehensible to the readers. Please send this article for proofreading before publication. Please note the differences between US/UK spelling.
e.g
line 57: "of"
line 85: "input"
line 138: Morocco
line 153: another result
line 235: ?
Response: The whole manuscript has been carefully checked and revised in order to correct English language errors.
- There is a jump between numbering (2) Results and Discussion and (4) Materials and Methods.
Response: The numbering of sections and subsections has been corrected.
3. Section 4.2 was copied from one of the authors' other publications.
Response: Section 4.2 has been revised.
4. The abbreviations were not explained properly in the "Results and Discussion" because the "Materials and Methods" was placed at the end of the article.
Response: The manuscript has been carefully checked for abbreviations and the relevant explanation was added the first time the abbreviation appeared in the text. The “Materials and methods” section in placed after the “Results and Discussion” as this is the manuscript layout required by the Molecules journal.
- There are missing references: line 272.
Response: The missing references 30 and 31 have been included in the manuscript.